# Non-Polio Enterovirus Surveillance in the Ural Federal District and Western Siberia, 2022: Is There a Need for a Vaccine?

**DOI:** 10.3390/vaccines11101588

**Published:** 2023-10-12

**Authors:** Tarek M. Itani, Vladislav I. Chalapa, Vasilii N. Slautin, Roman O. Bykov, Bolat S. Imangaliev, Polina K. Starikova, Aleksandr G. Sergeev, Aleksandr V. Semenov

**Affiliations:** 1Laboratory of Enteric Virus Infections, Federal Budgetary Institution of Science «Federal Scientific Research Institute of Viral Infections «Virome»», Federal Service for Surveillance on Consumer Rights Protection and Human Wellbeing, 620030 Yekaterinburg, Russia; chalapa_vi@niivirom.ru (V.I.C.); slautin_vn@niivirom.ru (V.N.S.); bykov_ro@niivirom.ru (R.O.B.); imangaliev_bs@niivirom.ru (B.S.I.); starikova_pk@niivirom.ru (P.K.S.); ald131250@yandex.ru (A.G.S.); semenov_av@niivirom.ru (A.V.S.); 2Department of Microbiology, Virology and Immunology, Ural State Medical University, 620109 Ekaterinburg, Russia; 3Institute of Natural Sciences and Mathematics, Ural Federal University Named after the First President of Russia B.N. Yeltsin, 620075 Ekaterinburg, Russia

**Keywords:** epidemiological surveillance, non-polio enteroviruses, hand-foot-and-mouth disease, herpangina, enteroviral meningitis, circulation, prevention, enterovius types, coxsackie A6, non-polio enterovirus vaccine

## Abstract

Human non-polio enteroviruses (NPEVs) are the etiological agents involved in most cases of hand-foot-and-mouth disease (HFMD), herpangina and aseptic meningitis. Information on the epidemiology profiles of NPEV in the Ural Federal District and Western Siberia is very limited, with no published data available. The aim of this study is to describe NPEV incidence in the Ural Federal District and Western Siberia among patients with different forms of non-polio enterovirus infections (NPEVIs) during 2022, stratified by age and clinical manifestations. A total of 265 samples that tested positive for NPEV using a polymerase chain reaction (PCR) were genotyped by semi-nested PCR for the VP1 gene. The results showed that 21 genotypes were identified among patients in this study. CVA6 was the most common genotype for HFMD. CVA6, along with CVA10, accounted for the majority of herpangina cases, while CVA9 was implicated in most meningitis cases. Sequence and phylogenetic analysis showed that nearly all of the CVA6 strains identified in this study displayed a close genetic relationship to strains identified in other cities in Russia and strains from China. NPEV surveillance allows for monitoring the circulation of clinically relevant genotypes, resulting in continuous data about NPEV epidemiology. This is important for improving case prevention, diagnosis and guiding clinical management.

## 1. Introduction

Non-polio enteroviruses (NPEVs) are non-enveloped viruses of the Picornaviridae family [1]. There are over a hundred known types of enteroviruses. Previously, they were classified as coxsackievirus A, coxsackievirus B, echovirus and a group of numbered enteroviruses (EV) [2]. Later, on the basis of molecular research, human enteroviruses were categorized into four species: Enterovirus A (coxsackievirus A 2–8, 10, 12, 14, 16, enterovirus 71, 76, 89–92, 114, and 119–125 types); Enterovirus B (coxsackievirus A9, coxsackievirus B1-B6, all echoviruses, enterovirus 69, 73–75, 77–88, 93, 97, 98, 100, 101, 106, 107, and 110–114); Enterovirus C (polioviruses, coxsackievirus A1, 11, 13, 15, 17–22, 24, enterovirus 95, 96, 99, 102, 104, 105, 109, 113, and 116–118) and Enterovirus D (enterovirus 68, 70, 94, 111, and 120) [3].

Most NPEV infections are asymptomatic. On the other hand, these viruses are known to cause a wide spectrum of clinical manifestations, such as stomatitis, acute respiratory infections, diarrhea, pharyngitis, herpangina, exanthema, aseptic meningitis, encephalitis, acute and chronic cardiac disease, pleurodynia, eye infections, and many others [4,5]. They can also cause mixed forms with exanthema (like herpangina with exanthema, pharyngitis with exanthema etc.). Generally, EV-A are mainly associated with herpangina and hand-foot-and-mouth disease (HFMD), EV-B with herpangina and viral meningitis or encephalitis, EV-C- with poliomyelitis, and EV-D with respiratory infections [5,6]. The immunity for enterovirus infection is serotype-specific [7].

NPEVs are also the main cause of aseptic meningitis. They make up from 85% to 95% of all aseptic meningitis cases with defined etiology [8]. Most of the enteroviral meningitis outbreaks around the world that were registered over the last few years were caused by different Enterovirus B types [9,10] and mainly by echoviruses [11,12,13]. This point is also supported by a meta-analysis on the global prevalence of enteroviruses, including distribution by types [14].

Because of numerous clinical manifestations of NPEV infections (NPEVIs), confirmation of the diagnosis can be important for reducing hospitalization, antibiotic use and additional diagnostic testing often performed to exclude or treat other conditions [15]. The diagnosis of NPEVIs in clinical samples is mainly conducted using molecular techniques because of their higher sensitivity, specificity and turn-around time when compared to classical virus isolation techniques [15,16].

It has been shown that a rise in NPEVI incidence is usually preceded by the emergence of a new genovariant of virus for this territory or the return of a long-absent type [17,18]. Thus, the surveillance of NPEVIs is of prime importance for Rospotrebnadzor (Russian Federal Service for Surveillance on Consumer Rights Protection and Human Wellbeing), which is responsible for carrying out the federal state sanitary and epidemiological surveillance and federal state monitoring in the field of consumer rights protection in the Russian Federation. The main objectives of type-based NPEV surveillance in the Ural Federal District and Western Siberia include: (i) to help public health practitioners determine long-term patterns of NPEV circulation; (ii) to support the recognition of disease outbreaks associated with circulating NPEV genotypes; (iii) to coordinate with the National Center for the Study of NPEVI in the Russian Federation (Blokhina Nizhny Novgorod Research institute of Epidemiology and Microbiology); and (iv) to monitor the detection of virus types associated with acute flaccid paralysis (AFP), like EV-D68 and EVA71.

With no published data available, the aims of our study are to conduct NPEV surveillance, to describe NPEVI incidence, and to characterize the main circulating genotypes in the Ural Federal District and Western Siberia during 2022. The results of NPEV typing are stratified by clinical manifestations and age.

## 2. Materials and Methods

### 2.1. Data Collection and Samples

The National Enterovirus Surveillance System has been collecting laboratory data and biological samples from enteroviral infections in the Ural Federal District and Western Siberia since 2017. The participating centers were asked to report NPEV detections monthly to the Ural–Siberian Regional Scientific and Methodological Center for the Study of Enteroviral Infections. Patients were recruited, after giving informed consent, from eleven federal subjects of the Ural Federal District (Sverdlovsk Oblast, Chelyabinsk Oblast, Tyumen Oblast, Khanty-Mansi Autonomous Okrug, Kurgansk Oblast and Yamalo-Nenets Autonomous Okrug) and Western Siberia (Tomsk Oblast, Omsk Oblast, Novosibirsk Oblast, Kemerovo Oblast and Altai krai) by the “Centers for Hygiene and Epidemiology of the Russian Federation of Rospotrebnadzor” (Figure 1). Samples were collected as part of the Russian state program for NPEV surveillance, following the guidelines of the National Center for the Study of NPEVI in the Russian Federation. PCR positive samples with high viral loads (PCR cycle-threshold less than 27 cycles) were collected by participating centers and were sent to our regional center. There was a limit of 40 samples set for each participating center, although this limit was increased for the Sverdlovsk Oblast and the Khanty-Mansi Autonomous Okrug (71 and 56 samples, respectively) because of numerous outbreaks in these two federal districts. According to national regulations, the use of anonymous samples and data from state epidemiological surveillance does not require informed consent.

For laboratory studies, fecal samples were taken from patients with confirmed non-polio enterovirus infections for analysis during the first day of hospitalization from disposable diapers or from disposable plastic bags placed in a vessel or other container for defecation for older children. Fecal samples were stored as fecal suspensions, where 0.1–0.2 gr. of feces was dissolved in 1–1.5 mL of physiological saline (0.9% NaCl) and stored at −20 °C. For specific clinical manifestations, other biological samples were taken: cerebrospinal fluid (CSF) for patients with enteroviral meningitis; nasopharyngeal swabs and smears from the lesion sites of patients with exanthema of the mucous membranes; and skin for patients with herpangina and vesicular stomatitis. All biological samples were frozen, and stored and transported at −20 °C.

### 2.2. RNA Extraction and Molecular Detection

First, stool samples were centrifuged at 13,000× *g* for 2 min at room temperature to remove the solids. Then, total RNA was extracted from 200 μL of NPEV-positive samples using the RIBO-prep kit (InterLabService Ltd., Moscow, Russia) according to the manufacturer’s recommendations. The extracts were eluted in 50 μL of elution buffer and immediately used for molecular testing. The presence of NPEV in the samples was confirmed by a real-time reverse transcriptase polymerase chain reaction (rRT-PCR) using a pan-enterovirus kit, AmpliSens^®^ Enterovirus-FL (FBSI Central Research Institute of Epidemiology Rospotrebnadzor, Moscow) according to the manufacturer’s instructions. The experiments were performed using the ABI StepOne Plus Real-time PCR system (Applied Biosystems, Waltham, MA, USA).

### 2.3. Sequencing of the VP1 Gene

The identification of enteroviral type was performed by RT- semi-nested PCR and partial sequencing of the VP1 genomic region, using oligonucleotide primers and a protocol developed by Nix et al. [19], with minor modifications. Briefly, isolated RNA was used in a reverse transcription reaction for the synthesis of cDNA using the REVERTA-L-100 kit (InterLabService Ltd., Moscow, Russia). Partial VP1 genomic region of the synthesized cDNA was amplified as previously described [19]. The reaction products were separated and visualized on 1.0% agarose gel and extracted by using a PureLink™ PCR Purification Kit (Invitrogen, Waltham, MA, USA). The purified products were sequenced in both directions (forward and reverse) using the deoxy sequencing Sanger method on an automatic genetic analyzer 3130 Genetic Analyzer (Thermo Fisher Scientific, Waltham, MA, USA), following the manufacturer’s protocol.

### 2.4. Genotyping and Phylogenetic Analysis

The generated sequences were manually curated using Chromas 2.6.6 (Technelysium, Pty, Ltd., Queensland, Australia, version 2.6.6), and the consensus sequences were analyzed using the online Basic Local Alignment Search (BLAST) program (http://blast.ncbi.nlm.nih.gov/Blast.cgi (accessed on 1 February 2023)) to compare the homology sequences with the genome sequences of NPEVs from the international genetic database, NCBI GenBank. In addition, genotyping was confirmed using the online RIVM program (https://www.rivm.nl/mpf/typingtool/enterovirus/, accessed on 2 February 2023). From the total number of obtained sequences, 40 nucleotide sequences of NPEVs were deposited in the GenBank database: OP762462-OP762500.

The phylogenetic trees were constructed with the Molecular Evolutionary Genetics Analysis (MEGA7) software, version 7.0, Pennsylvania, USA (http://megasoftware.net/ (accessed on 3 March 2023)) using Kimura’s two-parameter evolutionary model (K80) and the Neighbor Joining algorithm with 2000 bootstrap replicates [20]. For phylogenetic analysis, Coxsackie A6 sequences were included. Clustering was performed over the VP1 genome region.

### 2.5. Statistical Analysis

Contingency tables were analyzed using Pearson’s chi-squared test and Fisher’s exact test (two-tailed). Differences were considered statistically significant at *p* < 0.05.

## 3. Results

### 3.1. Epidemiology and Patient Demographics

In 2022, 2714 cases of NPEVIs were registered in the Ural Federal District and Western Siberia, and the incidence rate was 11.91 per 100 thousand inhabitants. The long-term dynamics of the incidence of NPEVIs was characterized by a decrease in 2020, followed by an increase in 2021 and a recovery to pre-pandemic values in 2022 (Figure 2). The seasonal increase in incidence was registered in August–November, which is typical for a temperate climate and is consistent with retrospective data. The prevailing clinical form was HFMD/other rash (73.3%), and the proportion of meningitis/meningoencephalitis was 15.5%, with mild clinical forms (including enterovirus fever) accounting for 3.5%, and other forms (including gastrointestinal forms), 7.7%.

A total of 265 samples of clinical material from all 11 supervised regions were received and processed between January 2022 and December 2022 (Table 1). The predominant type of specimen material was feces (57.7%, 153/265), followed by oropharyngeal smears (37.4%, 99/265), cerebrospinal fluid (4.5%, 12/265) and vesicle discharge (0.4%, 1/265). The seasonality of sampling corresponded with the seasonality of the disease (Figure 2). The majority of the examined patients (95.5%, 253/265) were children. The proportions of the age groups were: 0–2 years, 36.2% (96/265); 3–6 years, 39.2% (104/265); and 7–17 years, 20.0% (53/265). The median age of the patients was 3 years (IQR 2–6), with a gender ratio of 1.00 (female):1.19 (male). Patients with HFMD were the most prevalent among those examined (43.0%, 114/265).

The largest number of samples were obtained from two regions with the highest incidence rates: Sverdlovsk Oblast (26.8%, n = 71) and Khanty-Mansi Autonomous Okrug (21.1%, n = 56), followed by Yamalo-Nenets Autonomous Okrug (12.1%, n = 32), Kurgan Oblast (9.4%, n = 25), Tomsk Oblast (7.5%, n = 20), Tyumen Oblast (5.7%, n = 15), Kemerovo Oblast (4.9%, n = 13), Chelyabinsk Oblast (4.5%, n = 12), Omsk Oblast (3.4%, n = 9), Novosibirsk Oblast (2.3%, n = 6) and Altai Krai (2.3%, n = 6).

### 3.2. Molecular Typing

Enteroviruses were successfully genotyped in 209 samples (78.9%) and a total of 21 different genotypes were detected (Table 2 and Table 3). Enteroviruses from different types of biomaterial were genotyped with comparable frequencies: 82% for pharyngeal smears (81/99), 77% for feces (118/153) and 75% for cerebrospinal fluid (9/12); the differences found were not statistically significant. EV-A types were identified in 144 samples (68.9%), and the most predominant genotype was CVA6 (40.7%), followed by CVA16 (10.5%) and CVA10 (7.7%) (Table 2). EV-B types were identified in 63 samples (30.1%), and the most common representative of the species was CV-A9 (9.6%), followed by E6 (5.3%) and CV-B2 (4.8%). Only two EV-C types were found (1.0%).

The structure of clinical forms caused by EV-A and EV-B was not the same: representatives of the species EV-B caused meningitis more often, while EV-A caused mainly HFMD and herpangina (Fisher’s exact test, *p* < 0.000001). The main causative agents of meningitis were CV-A9 (28%, 11/39), CV-A6 (23%, 9/39), CV-B2 (21%, 8/39) and E6 (18%, 7/39). Echovirus E30 from circulation was found in only one sample.

EV-A were more prevalent in infants under the age of three, contrasting with higher proportions of EV-B in older children and adults (Table 3). More than 75% of the samples (200/265) were received from children less than 6 years of age. CVA6 was the most common genotype in these samples (70 out of the 158 typed samples, 44.3%), followed by CVA16 (21/158, 13%) and CVA9 (15/158, 9.5%). A total of 20% of the received samples (53/265) belonged to children from 7 to 17 years of age. CVA6 was the predominant type (60.4%, 7/43), followed by E9 (16.2%, 7/43). Only 4,5% of samples were received from adults with no predominant type.

### 3.3. Sequence and Phylogenetic Analysis of Enterovirus Genotypes

To conduct the phylogenetic analysis using random sampling, positive samples from patients infected with CVA6 from the regions of the Ural Federal District and Western Siberia were included in the analysis. During 2022, 85 strains of CVA6 were identified. The phylogenetic tree is based on a partial sequencing of the VP1 genomic region.

Phylogenetic analysis of CVA6 genovariants (13 isolates) showed that most of the identified enteroviruses circulated in the studied regions for some years. As shown in Figure 3, the strains of CVA6 are divided into two clusters. Presented CVA6 strains were clustered with strains that circulated in China during 2021, rather than with ones previously detected in Russia, confirming its recent cross-border transportation.

## 4. Discussion

There is limited information on circulating types and associated clinical phenotypes in the Ural Federal District and Western Siberia. In this study, the non-polio enterovirus surveillance and characterization of the main circulating types was conducted in 2022. The percentage of specimen testing positive peaked in the summer or early fall of 2022. HFMD was the most common form of NPEVI in the Ural Federal District and Western Siberia. Over the past six years of surveillance, our region had numerous HFMD outbreaks, and the most recent one was in the town of Serov, a northern town of the Sverdlovsk region, with three-hundred and forty-six registered cases of HFMD and two cases of NPEV meningitis from July to September 2022, and more than ninety registered outbreaks. CVA6, CVA16 and CVA9 were the most frequently isolated genotypes during this period. The results of our research indicate the clear leadership of CVA6 in HFMD and EVB in meningitis. The prevalent types of NPEV among children below the age of 6 years were CVA6, CVA16 and CVA9. Among 7- to 17-year-old children, the prevalent types were CVA6 and Echovirus 7. EV-A were more prevalent in infants under the age of three, contrasting with higher proportions of EV-B in older children and adults (Table 3), as previously described [21].

Our analysis of NPEV etiology in the region of the Ural Federal District and Western Siberia allowed us to show that CVA6 played a leading role in development of localized or “minor” forms such as HFMD and herpangina. Many published studies from China have reported CVA6 as the most common genotype in HFMD [22,23]. In a Chinese study by Xia et al., the authors found that CVA6 shows the greatest transmission ability among these three pathogens (CVA6, CVA16 and EV-A71), while EV-A71 exhibits the weakest transmission ability [24]. Moreover, different climate conditions in various parts of China may have decreased the basic reproduction number for CVA16 and EV-A71 [24]. CVA16 and EV-A71 peaked before 2018 in China, and a large part of the population has been immune to these two pathogens. In another study from China by Han et al., the authors observed the predominance of EV-A71 and CVA16 during 2009–2018, with a significant decrease in EV-A71 after the introduction of the EV-A71 vaccination [25]. In general, according to a recent meta-analysis by Brouwer et al., CVA6 is the most prevalent NPEV globally [14]. CVA6 has attracted considerable attention and its association with HFMD outbreaks was first reported in Finland in 2008 [26], then later in Taiwan [27], and as part of a large outbreak in Singapore [28]. CVA6 infections cause a range of clinical issues, including severe or atypical HFMD and herpangina, although with fewer complications. Concerning herpangina cases, CVA6, CVA16 and CVA10 were predominant (74.2% of all cases) when compared to other genotypes. Peng et al. in 2015 reported that CVA2 accounted for the majority of herpangina cases [22]. In contrast, surveillance studies in Vietnam and Korea found that CVA6 and CVA16 were less prevalent than EVA71 and proposed this agent as a primary target for vaccine development [29,30].

Severe NPEVIs in the regions of the Ural Federal District and Western Siberia in 2022 were mainly caused by Enterovirus B (aseptic meningitis—75%). The most isolated pathogen in aseptic meningitis cases was the CVA9 type (27.5%).

We reported only eight cases of EVA71 in 2022; all were mild and without neurologic manifestations. In a Senegalese study by Ndaye et al. in 2022, a low number of isolates tested positive for EVA71 (9 out of 521 NPEV isolates) [31]. While a recent study from South Vietnam reported the predominance of EVA71 as the etiological agent of HFMD, AFP and enteroviral meningitis [32]. EVA71 is responsible for large outbreaks in South East Asian countries, while in Europe it is mainly associated with CNS symptoms [33].

E30 disappeared from circulation after many years of active circulation in the Russian Federation ([21]; A. Sergeev, personal communication, September 8, 2023) and in European countries like Germany [33]. Intensified hygiene, wearing masks and physical distancing, as well as the closure of daycares and schools during the COVID-19 pandemic, may be responsible for the dip in NPEVIs due to E30.

No EV-D68 type strains were isolated in this study, despite it being the most frequently reported type during 2014–2016 in the United States of America (55.9% of all cases) [18]. EV-D68 is far more likely to be detected in respiratory specimens than in stool specimens. Molecular typing has increased in recent years because it saves time compared to cell culture and has become affordable [33]. However, typing EV directly from clinical material remains challenging due to low viral loads in specimen like CSF [33].

Although the sampling in this study peaked during July–August, in parallel with the increase in the incidence of NPEVIs, samples were not evenly received from all participating centers. Because samples with low viral loads are hard to type using sanger sequencing, they were either not sent to our regional center or were not successfully typed.

Phylogenetic analysis compared the CVA6 strains that circulated in the Russian Federation and neighboring countries since 2017, and the CVA6 strains from our study. The CVA6 strains from the Sverdlovsk region and Khanty-Mansi Autonomous Okrug were genetically close to an isolate strain detected in China in 2021. Other strains isolated from the Chelyanbinsk, Yamalo-Nenets and Novosibirsk regions clustered together and were genetically related to a strain isolated from the Khanty-Mansi Autonomous Okrug in 2021.

At present, only monovalent inactivated whole virus vaccines containing EV-71 that have been approved for usage or are in the advanced stages of research are well-recognized. Notable among these inactivated whole virus vaccines are EVA71 from China (containing three vaccines with virus genotype C4 which exhibit remarkable efficacy against HFMD) and the EV71vac vaccine from Taiwan, developed based on enterovirus genotype B4. Monovalent vaccines comprising inactivated CVA16, CV-A10 or CV-A6 virions have showcased substantial efficacy, albeit they have only been studied in animal models [34]. An additional promising category of vaccines is represented by recombinant subunit vaccines, which hold potential in the treatment of enteroviruses and have demonstrated significant effectiveness in pre-clinical studies. Virus-like particles (VLPs) have exhibited robust titers of neutralizing antibodies, inducing a high antigen-specific B cell response comparable to that provoked by inactivated vaccines produced through diverse biosystems [35,36]. Furthermore, multivalent vaccines present themselves as an alternative and promising approach. The bivalent EV-A71\CV-A16 vaccine, evaluated using rhesus macaque models, demonstrated excellent antiviral efficacy and the ability to shield against viral infections without eliciting side-effects [35,36]. The trivalent vaccine formulations EV-A71\CV-A16\CV-A6 and CV-A6\CV-A10\CV-A16 also displayed high effectiveness in inducing immunization and providing protection against the development of severe complications [37]. The tetravalent vaccine, CV-A10\EV-A71\CV-A16\CV-A6, exhibited sustained immunization and effectiveness, mirroring the performance of the monovalent vaccines [38]. Given the serious and potentially life-threatening complications associated with hand-foot-and-mouth disease, the necessity of the prioritization of vaccine development is evident. Presently, inactivated vaccines demonstrate considerable efficacy, persistent immunogenicity and acceptable safety profiles within the vaccinated population [37]. Our results suggest that only polyvalent vaccines, including CV-A6\CV-A10\CV-A16, may be viable options for enterovirus control in Ural and Western Siberia. The only approved EVA71 vaccines are irrelevant for this study population.

## 5. Conclusions

There was a low number of samples and positivity rate observed in 2020–2021 in the Ural Federal District and Western Siberia. This was obviously due to the COVID-19 pandemic. Because of lockdowns and quarantine, children did not attend schools and kindergartens, which meant that the spread of NPEV was dramatically hindered. Another reason for this drop was the pressure that COVID-19 unleashed upon the world’s healthcare systems was so high that there were problems with surveillance.

The main goal of the NPEV surveillance system is to diminish the incidence of NPEVI and the economic burden of this infection in all possible ways. A timely and robust system has the potential to inform disease prevention strategies by supporting the recognition of outbreaks and guiding the development of new diagnostic tests and interventions. This will make it possible to foresee the development of outbreaks and to take timely preventive measures based on new technological solutions, especially in the field of clinical management and vaccination.

## Figures and Tables

**Figure 1 vaccines-11-01588-f001:**
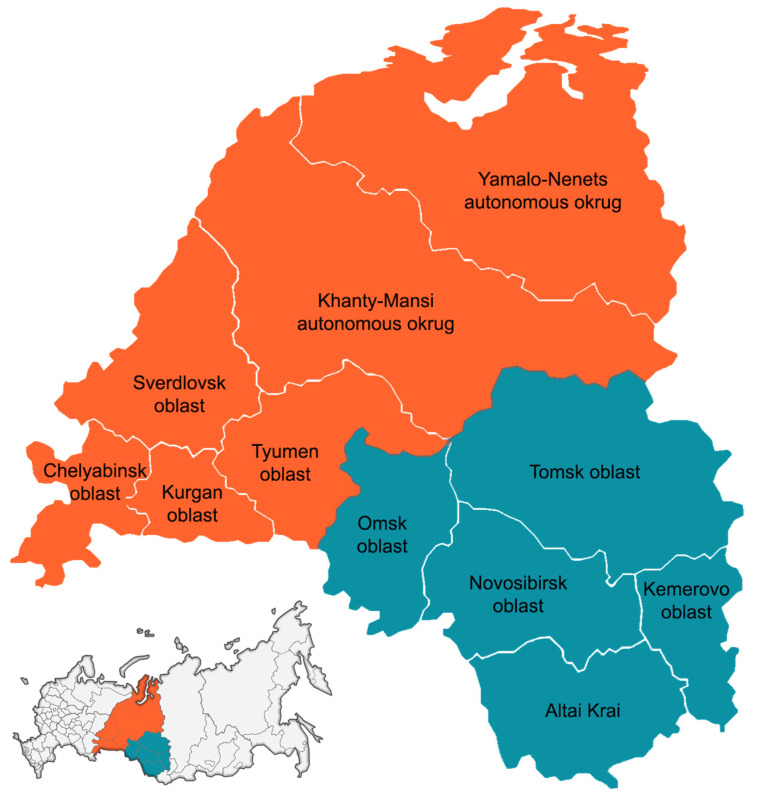
Federal subjects of the Ural Federal District and Western Siberia participating in this study.

**Figure 2 vaccines-11-01588-f002:**
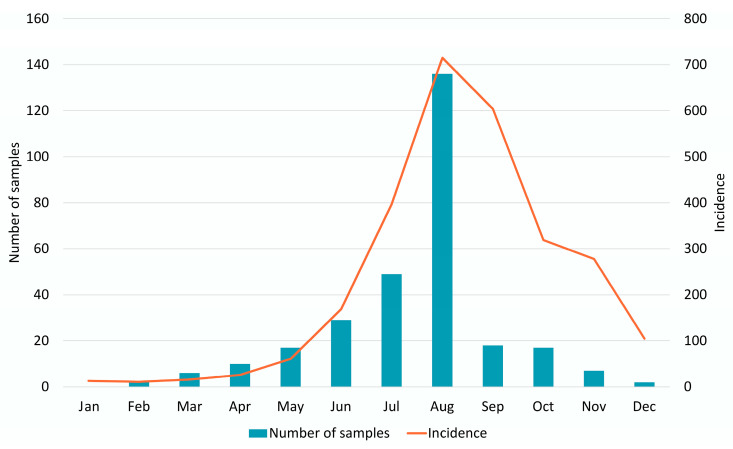
Monthly distribution of NPEVI incidence and sampling from patients, 2022.

**Figure 3 vaccines-11-01588-f003:**
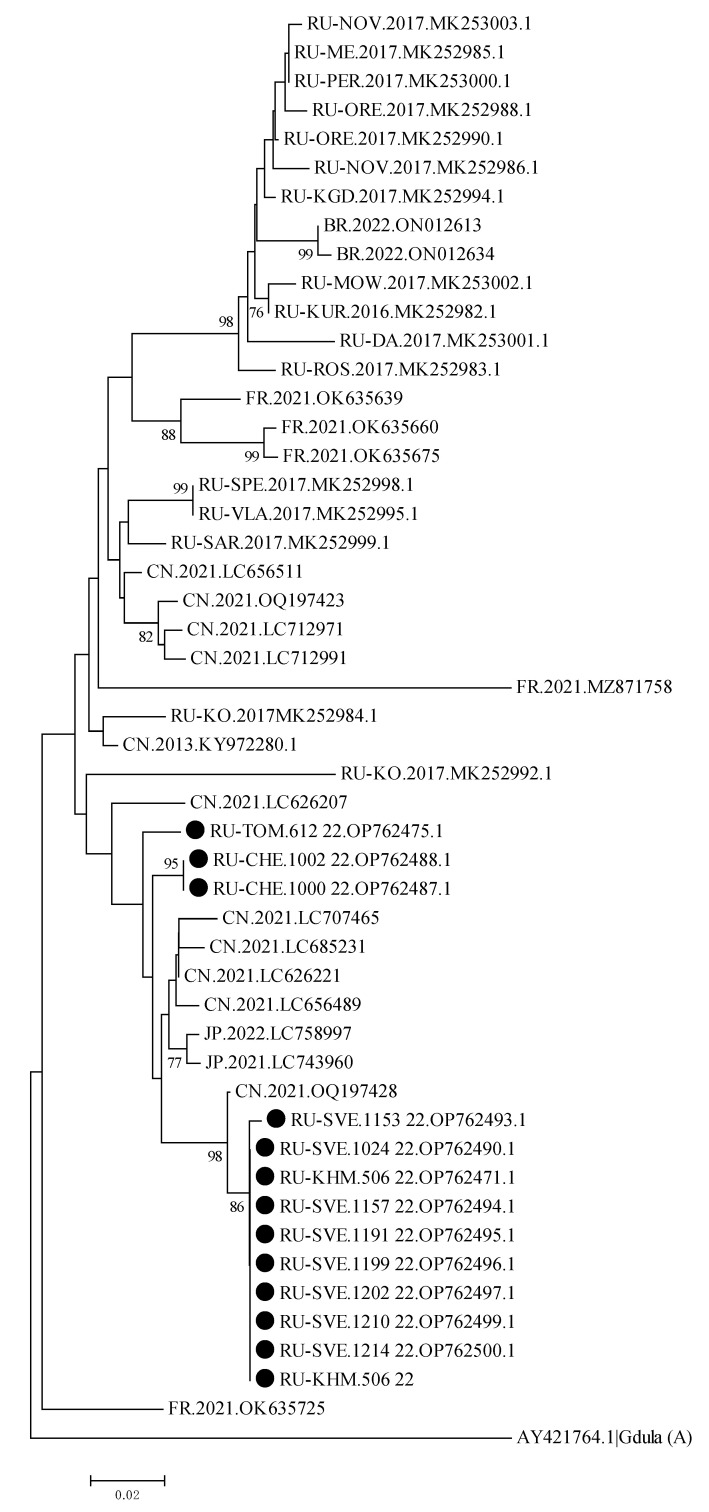
Phylogenetic tree of 13 Coxsackievirus A6 strains detected in the Ural Federal District and Western Siberia in 2022 (dotted). Bootstrap values lower than 70 were hided. Strains from Russian federal subjects marked with corresponding ISO 3166-2 codes.

**Table 1 vaccines-11-01588-t001:** Distribution of NPEVI cases by age and clinical forms, number of cases (%).

Clinical Features	Age Groups				
	0–2 Years	3–6 Years	7–17 Years	Adults	Total
Meningitis/meningoencephalitis	10 (3.8)	19 (7.2)	14 (5.3)	5 (1.9)	48 (18.1)
Hand-foot-and-mouth disease	54 (20.4)	36 (13.6)	23 (8.7)	1 (0.4)	114 (43.0)
Herpangina	16 (6.0)	21 (7.9)	5 (1.9)	-	42 (15.8)
Gastrointestinal forms	4 (1.5)	2 (0.8)	1 (0.4)	1 (0.4)	8 (3.0)
Exanthematous fever	5 (1.9)	3 (1.1)	-	1 (0.4)	9 (3.4)
Other infections	7 (2.6)	23 (8.7)	10 (3.8)	4 (1.5)	44 (16.6)
**Total**	**96 (36.2)**	**104 (39.2)**	**53 (20.0)**	**12 (4.5)**	**265 (100)**

**Table 2 vaccines-11-01588-t002:** Species and types of the NPEV genotypes according to clinical forms, n (%).

Species/Types	Meningitis/Meningoencephalitis	Hand-Foot-and-Mouth Disease	Herpangina	Gastrointestinal Forms	Exanthematous Fever	Other Infections	Total, n (%)
Coxsackievirus A2	-	1 (0.5)	-	-	-	-	1 (0.5)
Coxsackievirus A3	-	1 (0.5)	-	-	-	-	1 (0.5)
Coxsackievirus A4	-	2 (1.0)	2 (1.0)	-	-	-	4 (1.9)
Coxsackievirus A5	-	1 (0.5)	-	-	-	-	1 (0.5)
Coxsackievirus A6	9 (4.3)	53 (25.4)	9 (4.3)	-	5 (2.4)	9 (4.3)	85 (40.7)
Coxsackievirus A8	-	3 (1.4)	2 (1.0)	-	-	1 (0.5)	6 (2.9)
Coxsackievirus A10	-	1 (0.5)	9 (4.3)	1 (0.5)	2 (1.0)	3 (1.4)	16 (7.7)
Coxsackievirus A16	-	8 (3.8)	8 (3.8)	-	1 (0.5)	5 (2.4)	22 (10.5)
Enterovirus A71	-	5 (2.4)	1 (0.5)	-	1 (0.5)	1 (0.5)	8 (3.8)
**Enterovirus A**	**9 (4.3)**	**75 (35.9)**	**31 (14.8)**	**1 (0.5)**	**9 (4.3)**	**19 (9.1)**	**144 (68.9)**
Coxsackievirus A9	11 (5.3)	5 (2.4)	1 (0.5)	-	-	3 (1.4)	20 (9.6)
Coxsackievirus B2	8 (3.8)	1 (0.5)	1 (0.5)	-	-	-	10 (4.8)
Coxsackievirus B5	-	-	-	1 (0.5)	-	1 (0.5)	2 (1.0)
Echovirus E5	-	-	-	1 (0.5)	-	-	1 (0.5)
Echovirus E6	7 (3.3)	4 (1.9)	-	-	-	-	11 (5.3)
Echovirus E7	-	1 (0.5)	-	1 (0.5)	-	-	2 (1.0)
Echovirus E9	2 (1.0)	1 (0.5)	-	-	-	6 (2.9)	9 (4.3)
Echovirus E11	2 (1.0)	-	-	-	-	1 (0.5)	3 (1.4)
Echovirus E25	-	-	2 (1.0)	-	-	2 (1.0)	4 (1.9)
Echovirus E30	-	1 (0.5)	-	-	-	-	1 (0.5)
**Enterovirus B**	**30 (14.4)**	**13 (6.2)**	**4 (1.9)**	**3 (1.4)**	**-**	**13 (6.2)**	**63 (30.1)**
Coxsackievirus A1	-	-	-	-	-	1 (0.5)	1 (0.5)
Coxsackievirus A22	-	-	-	1 (0.5)	-	-	1 (0.5)
**Enterovirus C**	**-**	**-**	**-**	**1 (0.5)**	**-**	**1 (0.5)**	**2 (1** **.0** **)**
**Total, n (%)**	**39 (18.1)**	**88 (43.0)**	**35 (15.8)**	**5 (3.0)**	**9 (3.4)**	**33 (16.6)**	**209 (100)**

**Table 3 vaccines-11-01588-t003:** Species and types of the NPEV genotypes according to age groups of patients, n (%).

Species/Types	0–2 Years	3–6 Years	7–17 Years	Adults	Total, n (%)
Coxsackievirus A2	1 (0.5)	-	-	-	1 (0.5)
Coxsackievirus A3	-	1 (0.5)	-	-	1 (0.5)
Coxsackievirus A4	1 (0.5)	3 (1.4)	-	-	4 (1.9)
Coxsackievirus A5	1 (0.5)	-	-	-	1 (0.5)
Coxsackievirus A6	38 (18.2)	32 (15.3)	14 (6.7)	1 (0.5)	85 (40.7)
Coxsackievirus A8	3 (1.4)	1 (0.5)	2 (1)	-	6 (2.9)
Coxsackievirus A10	7 (3.3)	5 (2.4)	3 (1.4)	1 (0.5)	16 (7.7)
Coxsackievirus A16	7 (3.3)	14 (6.7)	1 (0.5)	-	22 (10.5)
Enterovirus A71	4 (1.9)	2 (1)	2 (1)	-	8 (3.8)
**Enterovirus A**	**62 (29.7)**	**58 (27.8)**	**22 (10.5)**	**2 (1)**	**144 (68.9)**
Coxsackievirus A9	4 (1.9)	11 (5.3)	5 (2.4)	-	20 (9.6)
Coxsackievirus B2	3 (1.4)	4 (1.9)	1 (0.5)	2 (1)	10 (4.8)
Coxsackievirus B5	1 (0.5)	-	-	1 (0.5)	2 (1)
Echovirus E5	1 (0.5)	-	-	-	1 (0.5)
Echovirus E6	2 (1)	3 (1.4)	5 (2.4)	1 (0.5)	11 (5.3)
Echovirus E7	1 (0.5)	1 (0.5)	-	-	2 (1)
Echovirus E9	-	1 (0.5)	7 (3.3)	1 (0.5)	9 (4.3)
Echovirus E11	-	1 (0.5)	2 (1)	-	3 (1.4)
Echovirus E25	-	3 (1.4)	-	1 (0.5)	4 (1.9)
Echovirus E30	-	-	1 (0.5)	-	1 (0.5)
**Enterovirus B**	**12 (5.7)**	**24 (11.5)**	**21 (10)**	**6 (2.9)**	**63 (30.1)**
Coxsackievirus A1	1 (0.5)	-	-	-	1 (0.5)
Coxsackievirus A22	-	1 (0.5)	-	-	1 (0.5)
**Enterovirus C**	**1 (0.5)**	**1 (0.5)**	**-**	**-**	**2 (1)**
**Total, n (%)**	**75 (36.2)**	**83 (39.2)**	**43 (20.0)**	**8 (4.5)**	**209 (100)**

## Data Availability

Not applicable.

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
