# Peer review of "Non-Polio Enterovirus Surveillance in the Ural Federal District and Western Siberia, 2022: Is There a Need for a Vaccine?"

_vaccines, 2023, doi:10.3390/vaccines11101588_

Round 1

Reviewer 1 Report

Overall, this is a very meaningful study with important implications for improving the preventive diagnosis and clinical guidance of human non-poliovirus in the Ural Federal District and Western Siberia. There are a few minor issues that need to be identified and corrected by the author before it can be published.

In line 53, the author describes that most of enteroviral meningitis outbreaks around the world that were registered during the last years were caused by different enterovirus B types. The literature the authors refer to is 10 and 20 years old respectively and does not draw a convincing conclusion. It is suggested that the author draw a definite conclusion according to the latest literature research.

In line 240, the author describes that many published studies from China have reported CVA6 as the most common genotype in HFMD. However, prior to 2018, EV-A71 and CVA16 were the two major pathogens causing HFMD in China. CVA6 is the main pathogen causing HFMD in China starting in 2018. Therefore, it is suggested that the author should draw a convincing conclusion based on the literature in recent years.

Author Response

For research article: Non-polio Enterovirus Surveillance in the Ural Federal District and Western Siberia, 2022: Is there a need for a vaccine?

Response to Reviewer 1 Comments

1. Summary

Thank you very much for taking the time to review this manuscript. Please find the detailed responses below and the corresponding revisions/corrections highlighted in red/in track changes in the re-submitted files.

2. Point-by-point response to Comments and Suggestions for Authors

-       Comment 1: In line 53, the author describes that most of enteroviral meningitis outbreaks around the world that were registered during the last years were caused by different enterovirus B types. The literature the authors refer to is 10 and 20 years old respectively and does not draw a convincing conclusion. It is suggested that the author draw a definite conclusion according to the latest literature research.

Response 1: We thank the reviewer for this very relevant comment. Therefore, we have updated our literature research and we have added three recent studies from Israel, Iran, and Cyprus. All these three studies have genotyped more enterovirus B types, especially Echovirus in meningitis cases and outbreaks. This point is also supported by a meta-analysis of global prevalence of enteroviruses, including its distribution by types (Brouwer, 2021), which has been added as well (lines 55-57)

The studies that have been added are:

-       Fratty IS, Kriger O, Weiss L, Vasserman R, Erster O, Mendelson E, Sofer D, Weil M. Increased detection of Echovirus 6-associated meningitis in patients hospitalized during the COVID-19 pandemic, Israel 2021-2022. J Clin Virol. 2023 May;162:105425. doi: 10.1016/j.jcv.2023.105425.

-        Farshadpour, F., Taherkhani, R. Molecular epidemiology of enteroviruses and predominance of echovirus 30 in an Iranian population with aseptic meningitis. J. Neurovirol. 27, 444–451 (2021). https://doi.org/10.1007/s13365-021-00973-1

-       Richter J, Tryfonos C, Christodoulou C. Molecular epidemiology of enteroviruses in Cyprus 2008-2017. PLoS One. 2019 Aug 8;14(8):e0220938. doi:10.1371/journal.pone.0220938.

-       Brouwer, L.; Moreni, G.; Wolthers, K.C.; Pajkrt, D. World-Wide Prevalence and Genotype Distribution of Enteroviruses. Viruses 2021, 13, 434, doi:10.3390/v13030434.

Comment 2: In line 240, the author describes that many published studies from China have reported CVA6 as the most common genotype in HFMD. However, prior to 2018, EV-A71 and CVA16 were the two major pathogens causing HFMD in China. CVA6 is the main pathogen causing HFMD in China starting in 2018. Therefore, it is suggested that the author should draw a convincing conclusion based on the literature in recent years.

Response 2:  Thank you for pointing this out. We agree with this comment. We thank the reviewer for this very important comment. We found a very interesting study by Xia et al, explaining why CVA6 replaced the previous dominating genotypes. In addition, we found a study suggesting the decrease in EV-A71 following the introduction of EV-A71 vaccination in China. The following paragraph has been added to the text (lines 252-262).

“In a Chinese study by Xia et al, authors have found that CVA6 shows the greatest transmission ability among these three pathogens (CVA6, CVA16, and EV-A71), while EV-A71 exhibits the weakest ability of transmission. Moreover, different climatic conditions in various parts of China, might have decreased the basic reproduction number for CVA16 and EV-A71”. CVA16 and EV-A71 peaked before 2018 in China, and a big part of the population has been immune to these two pathogens.

In another study from China by Han et al, authors observed the predominance of EV-A71 and CVA16 during 2009-2018 with significant decreasing of EV-A71 proportion after the introduction of EV-A71 vaccination. In general, according to a recent meta-analysis by Brouwer et al, CVA6 is most prevalent NPEV globally.”

The studies that have been added are:

-       Xia F, Deng F, Tian H, He W, Xiao Y, Sun X. Estimation of the reproduction number and identification of periodicity for HFMD infections in northwest China. J Theor Biol. 2020 Jan 7;484:110027. doi: 10.1016/j.jtbi.2019.110027. Epub 2019 Sep 27. PMID: 31568791.

-       Han, Y.; Chen, Z.; Zheng, K.; Li, X.; Kong, J.; Duan, X.; Xiao, X.; Guo, B.; Luan, R.; Long, L. Epidemiology of Hand, Foot, and Mouth Disease Before and After the Introduction of Enterovirus 71 Vaccines in Chengdu, China, 2009-2018. Pediatr Infect Dis J 2020, 39, 969–978, doi:10.1097/INF.0000000000002745.

-       Brouwer, L.; Moreni, G.; Wolthers, K.C.; Pajkrt, D. World-Wide Prevalence and Genotype Distribution of Enteroviruses. Viruses 2021, 13, 434, doi:10.3390/v13030434.

Reviewer 2 Report

I've been invited to review a very interesting paper from Russian Federation (Ural Region) entitled: 

Non-polio Enterovirus Surveillance in the Ural Federal District and Western Siberia, 2022: Is there a need for a vaccine?

In this report, Itani et al. report on the results of the surveillance from a reference center on the Enterovirus infections from a central and quite wide region of Russian Federation.

The paper is well written and documented, and no significant flaws could be identified. The only shortcoming of this study is the lack of a preliminary step in this report. I will explain shortly:

Authors describe, accurately and in proper details, the characteristics of 265 samples. On the other hand, how these samples were actually collected, which characteristics led peripheral centers to collect and investigate these samples remains unclear. 

In other words, the only one but significant improvement needed by this paper is:

a) a preliminary section in the methods section where Authors describe how the samples were collected, following which guidelines, etc, in order to remove any potential selection bias;

b) some discussion about the limits of the sampling strategy based on the point a)

Author Response

For research article: Non-polio Enterovirus Surveillance in the Ural Federal District and Western Siberia, 2022: Is there a need for a vaccine?

Response to Reviewer 2 Comments

1. Summary

Thank you very much for taking the time to review this manuscript. Please find the detailed responses below and the corresponding revisions/corrections highlighted in red/in track changes in the re-submitted files.

2. Point-by-point response to Comments and Suggestions for Authors

-       Comment 1: a) a preliminary section in the methods section where Authors describe how the samples were collected, following which guidelines, etc, in order to remove any potential selection bias;

Response 1: We thank the reviewer for this remark. We have added a preliminary section, detailing which guidelines have been followed and how samples were collected. This section has been added (lines 93-101):

“Samples were collected following the guidelines of the National center for the study of NPEVI in the Russian Federation. PCR positive samples with high viral load (PCR cycle-threshold less than 27 cycles) were collected by participating centers and were sent to our regional center. There was a limit of 40 samples set for each participating center, although this limit was increased for the Sverdlovsk oblast and the Khanty-Mansi autonomous okrug (71 and 56 samples respectively), because of numerous outbreaks in these two federal districts”.

-       Comment 2: b) some discussion about the limits of the sampling strategy based on the point a)

Response 2:  We thank the reviewer for bringing this up. We understand the limitation of these types of studies, as usually we are very limited by the number of samples which are typable by sanger sequencing. We have added some discussion about the limits of the sampling strategy to the discussion section of the article (lines 288-291).

“Although the sampling in this study peaked during July-August in parallel with the increase in the incidence of NPEVI, samples were not evenly received from all participating centers. Because samples with low viral load are hard to type by using sanger sequencing, they were either not sent to our regional center or were not successfully typed.”